# Is Hypnotic Induction Necessary to Experience Hypnosis and Responsible for Changes in Brain Activity?

**DOI:** 10.3390/brainsci13060875

**Published:** 2023-05-29

**Authors:** Alejandro Luis Callara, Žan Zelič, Lorenzo Fontanelli, Alberto Greco, Enrica Laura Santarcangelo, Laura Sebastiani

**Affiliations:** 1Department of Information Engineering, University of Pisa, 56126 Pisa, Italy; alejandro.callara@unipi.it (A.L.C.); alberto.greco@unipi.it (A.G.); 2Department of Translational Research and New Technologies in Medicine and Surgery, University of Pisa, 56126 Pisa, Italy; zan.zelic@gmail.com (Ž.Z.); laura.sebastiani@unipi.it (L.S.); 3Department of Clinical and Experimental Medicine, University of Pisa, 56127 Pisa, Italy; lorenzo.fontanelli@outlook.it

**Keywords:** hypnotic induction, hypnosis, hypnotizability, expectation, beliefs, EEG

## Abstract

The relevance of formal hypnotic induction to the experience of trance and its neural correlates is not clear, in that hypnotizability, beliefs and expectation of hypnosis may play a major role. The aim of the study was assessing the EEG brain activity of participants with high (highs) or low hypnotizability scores (lows), aware of their hypnotizability level and informed that the session will include simple relaxation, formal hypnotic induction and neutral hypnosis. A total of 16 highs and 15 lows (according to the Stanford Hypnotic Susceptibility Scale, form A) were enrolled. Their EEGs were recorded during consecutive conditions of open/closed-eyes relaxation, hypnotic induction, neutral hypnosis and post hypnosis not interrupted by interviews. The studied variables were theta, alpha and gamma power spectral density (PSD), and the Determinism (DET) and Entropy (ENT) of the EEG signal Multidimensional Recurrence Plot (mRP). Highs reported significantly greater changes in their state of consciousness than lows across the session. The theta, alpha and gamma PSD did not exhibit condition-related changes in both groups. The Alpha PSD was larger in highs than in lows on midline sites, and the different sides/regions’ theta and gamma PSD were observed in the two groups independently from conditions. ENT showed no correlation with hypnotizability, while DET positively correlated with hypnotizability during hypnosis. In conclusion, the relevance of formal hypnotic induction to the experience of trance may be scarce in highs, as they are aware of their hypnotizability scores and expecting hypnosis. Cognitive processing varies throughout the session depending on the hypnotizability level.

## 1. Introduction

According to the American Psychological Association (APA), hypnosis is a state of focused attention, reduced contact with the environment and increased proneness to accept suggestions [1]. Despite being subjectively reported as different from normal wakefulness [2,3], hypnosis cannot be regarded as a distinct physiological state, as it cannot be defined independently from self-reports, unlike sleep stages [4]. Nonetheless, neuroimaging highlighted changes in the activity and connectivity of the Default, Salience and Executive Networks [5,6,7,8]. In particular, the hypnotic state has been associated with an increase in the functional connectivity between the anterior cingulate cortex and the dorsolateral prefrontal cortex, and with a reduced activation of the anterior part or of the entire Default Mode Network [8]. Specific neurotransmitters (dopamine, oxytocin, serotonin), in contrast, have been associated with hypnotizability rather than hypnosis [5]. In contrast, EEG studies based on the analysis of spectral frequencies [9,10,11,12,13] were often inconsistent with each other. Recent studies [14], in fact, revealed that changes in information processing during hypnosis are not necessarily associated with changes in local power and can be better measured with time series analyses [15]. Moreover, different connectivity could be responsible for different information processing not implying power changes, like the segregated mechanisms described by complexity indices during hypnosis [16]. The analyses of the EEG complexity [17,18,19], however, were often focused on the correlates of specific suggestions (like those administered during hypnotic assessment by standard scales) rather than solely on neutral hypnosis [13,20], which is the state following hypnotic induction not associated with any suggestion other than relaxation and pleasantness included in standard induction procedures [21].

The difficulty to physiologically characterize the experience of neutral hypnosis, which can be self-induced or achieved after hypnotic induction performed by a hypnotist [1], could be accounted for by the great individual variability, which makes it difficult to propose a unique model of “hypnotic state”. Indeed, spontaneous hypnotic states occur like normal fluctuations of the state of consciousness. Moreover, beliefs, the expectation of hypnosis and hypnotizability may show greater relevance than formal induction in the experience of trance [22,23,24,25,26,27]. In line with these observations, the physiological modifications induced by suggestions (requests to imagine sensory–motor and/or cognitive–emotional experiences different from the actual one and experiencing them as real) during hypnosis are not very different from those induced by the suggestions administered out of hypnosis [28,29,30].

In healthy participants with high hypnotizability scores (highs), informed about their level of hypnotizability and aware that the experimental session will include hypnotic induction, the effects of expectation, hypnotizability and beliefs may prevail over those of formal hypnotic induction, so that the EEG changes during the session might not be time-locked to it. Thus, the present study was aimed at identifying possible changes in the EEG brain activity of participants with high or low (lows) hypnotizability scores during a session including simple relaxation, formal hypnotic induction and neutral hypnosis.

## 2. Participants and Study Design

### 2.1. Participants

The study was conducted according to the Declaration of Helsinki and approved by the Bioethics Committee of the University of Pisa (n.17/2021). For the power spectral density (PSD) repeated measures ANOVA the minimum number of participants required to obtain α = 0.05 and (1-β) = 90% with the participants divided in 2 groups was 32, according to G^*^Power 3.1 analysis [31]. After their written informed consent, 198 healthy right-handed (Edinburgh Handedness Inventory, score > 16) students at the University of Pisa were submitted to hypnotic assessment by the Italian version of the Stanford Hypnotic Susceptibility Scale, form A (SHSS: A (range: 0–12) [21]). A total of 16 highs (SHSS score (mean ± sd): 10.18 ± 1.19; 9 females, age: 25.14 ± 3.82 years) and 16 lows (SHSS score: 0.20 ± 0.56; 7 females, age: 26.47 ± 4.82 years) were enrolled in the study. They had negative anamneses of medical, neurological and psychiatric disorders, attention/sleep disturbance and drug intake in the latest 6 months. Six highs and two lows reported an episodic practice of relaxation/meditation. Their EEG traces were analyzed in another study with different aims and methods [32].

### 2.2. Experimental Procedure

As previously described [32], on the day of the experimental session, the participants were invited to sit in an armchair and were prepared for EEG recording. The experimental sessions took place in the early afternoon, after the intake of the last meal and caffeine-containing beverages, in a sound- and light-attenuated, temperature-controlled (22 °C) room. They were invited to relax and informed that, after a few minutes, they will have to listen to a pre-recorded hypnotic induction (SHSS: A induction, modified by the direct request made to participants to close their eyes at a certain point of the session). The EEG traces were recorded during consecutive 3 min conditions: open-eyes relaxation (R_OE_), closed-eyes relaxation (R_CE_), earliest and latest part of the induction (IND_1_, IND_2_), neutral hypnosis (NH) and recovery after hypnosis (POST). The same experimenter invited all participants to relax (before R_OE_), to close their eyes (before R_CE_), to listen to the pre-recorded voice (after R_CE_) and to open their eyes after NH.

Finally, the participants scored the experienced change in their state of consciousness (∆SoC) throughout the session (“*How much did your state of consciousness change during the session?*”; range: min 0–max 10). Questions about the expectancy of ∆SoC during hypnosis were not asked before starting the session to prevent the participants from unintentionally reporting ∆SoC congruent with the declared expectancy, at its end, owing to expectation/motivation-induced placebo response [33,34]. Specific questionnaires (i.e., [2]) to characterize the state of consciousness were not administered after each condition to avoid altering the participants’ flow of the state of consciousness by interrupting the session. 

### 2.3. Signal Acquisition and Analysis

ECG and EEG signals were recorded with a telemetric Nautilus EEG system (g.tec, Schiedlberg). Leads were placed in F3, F7, C3, T7, P3, PO3, F4, F8, C4, T8, P4, PO4, Fz, Cz, Pz, Oz positions according to the International 10–20 System and referred to Cz. In addition, 2 electrodes were placed on the lateral cantus of the left eye and in correspondence to the left orbital ridge to detect eye movements. Another electrode, referred to Cz, was placed in at the level of the left shoulder for ECG acquisition. All impedances were kept below 5 kΩ. To clean the EEG signals from unwanted noise and artefacts, we used EEGLAB [35] and MATLAB 2020b [36] custom scripts. Specifically, all EEG signals were downsampled to a sampling frequency of 125 Hz, after applying a proper low-pass anti-aliasing filter. Then, a high-pass filter of 0.1 Hz was applied. Bad channels were removed through a semi-automatic procedure (for details see [32]) and successively recovered through a spline–spherical interpolation method. Signals were re-referenced to the numerical average of all channels and decomposed through independent component analysis (ICA) [37]. Finally, we reconstructed the signal on the scalp with the only contribution of independent components related to brain activity [38].

We estimated the PSD on cleaned EEG signals in a time-varying fashion. To this aim, we used a sliding window. Within each three-minute-long window (i.e., R_OE_, R_CE_, IND1, IND2, NH, POST), we estimated the PSD with the Welch method using five-second-long windows with an overlap of 80%. The PSD was then integrated in the canonical EEG frequency bands, i.e., delta (1–4 Hz), theta (4–8 Hz), alpha (8–12 Hz), beta (13–30 Hz), and gamma (30–45 Hz). 

In a similar fashion, we performed a time-varying Multidimensional Recurrence Quantification Analysis (MdRQA) of the EEG signal. QA measures have been used to characterize EEG signals because their evaluation does not require any assumptions on stationarity, length or noise of time series [38]. Here, we used one-minute-long sliding windows with no overlap. For each window, we used the MdRQA as implemented in [39]. This method extends the original implementation of the RQA [40] to multidimensional data and is thus especially suited for the analysis of the EEG signal. Particularly, it allows us to characterize the behavior of multivariate time-series that are the result of multiple interdependent variables, potentially exhibiting non-linear behavior over time [39,40,41]. To determine the MdRQA, it is necessary to first perform a phase-space reconstruction using the method of time-delayed embedding. This latter method needs two parameters to be estimated: the delay parameter τ, which is the lag at which the time series has to be plotted against itself, and the embedding dimension parameter D, where D − 1 is the number of times that the time series has to be plotted against itself using the delay τ [42]. Then, if these two parameters are known, one can reconstruct an approximation of the original phase-space dynamics from a one-dimensional time series [43]. The two methods that are typically used to estimate these parameters in one-dimensional time series are the Average Mutual Information (AMI) function and the False Nearest Neighbor (FNN) function. Here, we used their optimized versions for the multidimensional data implemented in [44]. Finally, we extracted Determinism (DET) and Entropy (ENT) from the obtained recurrence plots (RP) as the variables of interest. In fact, these variables had been studied in not-hypnotized highs, showing a significant group difference in DET at Pz [44,45]

### 2.4. Variables

Variables include scores of the change in the state of consciousness (∆SoC); EEG theta, alpha and gamma PSD, which are modulated during relaxation and hypnosis [9,46]; EEG Multidimensional Recurrence Plot (mRP); and DET and ENT, which have been found different between highs and lows during relaxation [44,45].

### 2.5. Statistical Analysis

One subject of the lows group was excluded from analysis owing to the many artefacts in the recorded signals. The Kolmogorov–Smirnov test was used to study the variables’ distribution. 

The Spearman correlation coefficient between the change in state of consciousness (∆SoC) and the SHSS scores was computed. Log transformed theta, alpha and gamma hemispheric PSD were analyzed through a design of 2 Groups (highs, lows) × 2 Sides (left, right) × 3 Regions [(frontal (F), central-temporal (C-T), parieto-occipital (P-O) × 6 Conditions (R_OE_, R_CE_, IND_1_, IND_2_, NH, Post). Midline sites (Fz, Cz, Pz) were analyzed for the same frequency bands and time domain indices through a 2 Groups × 3 Regions (Fz, Cz, Pz) × 6 Conditions (R_OE_, R_CE_, IND_1_, IND_2_, NH, Post) repeated measures ANOVA. The Greenhouse–Geisser ε correction for non-sphericity was applied when necessary. Post hoc comparisons were conducted by paired t-tests between conditions and regions. The level of significance was set at *p* = 0.05.

To observe group and condition significant effects/interactions for DET (*d* = 0.203) and ENT (*d* = 0.127), a larger sample (N ≥ 96 for DET, N ≥ 86 for ENT) would be required. Since we could not increase the sample size within the time approved for the study by the Bioethics Committee, we limited the analysis to the bivariate Spearman correlations between SHSS scores and DET/ENT. 

## 3. Results

None of the participants showed EEG sleep episodes during the session, but two lows reported sleepiness (∆SoC = 7). The mean values of ∆SoC, significantly higher in highs (mean ± SD; 6.90 ± 1.17) than in lows (1.57 ± 2.21), as well as the satisfaction of their last night’s sleep, similar in highs and lows, were reported in [32]. A significant Spearman correlation between SHSS scores and the ∆SoC experienced over the entire session was observed (ρ = 0.810, *p* < 0.001). 

### 3.1. EEG Spectral Analysis

#### 3.1.1. Theta PSD

At hemispheric level, decomposition of the Side × Group (*F*(1, 29) = 13.15, *p* = 0.0001, η^2^ = 0.312, α = 0.939) revealed significant side differences in lows (right > left, *F*(1, 14) = 24.06, *p* = 0.0001) and no significant difference between groups on each side (Figure 1a).

Significant Side × Region (*F*(2, 28) = 26.14, *p* = 0.001, η^2^ = 0.474, α = 1.00) and Region × Condition (*F*(10, 140) = 2.17, *p* = 0.039, η^2^ = 0.070, α = 0.908) interactions were also observed. Decomposition of the Side × Region interaction (Appendix A) showed significant differences between all regions within the same side and between the corresponding regions of the two sides, except for the similar right and left P-O regions. Decomposition of the Region × Condition interaction indicated significant differences between almost all regions, with the F and P-O PSD larger than the C-T PSD in all conditions, and the F PSD larger than the P-O PSD only during Roe and IND1 (Appendix A).

At midline sites, a significant **Region × Group** interaction *F*(2, 58) = 5.84, *p* = 0.005, η^2^ = 0.168 α = 0.855) showed a Region effect in both highs (*F*(2, 30) = 38.61, *p* = 0.0001) and lows (*F*(2, 28) = 55.36, *p* = 0.0001, η^2^ = 0.798, α = 1.00). Highs exhibited a lower theta PSD than lows (Figure 1b) at the frontal region (*t*(29) = 2.58, *p* = 0.015). Moreover, in highs, the frontal PSD was larger than the central-temporal (*t*(15) = 7.41, *p* = 0.0001) and parieto-occipital PSD (*t*(15) = 9.59, *p* = 0.0001), whereas there was no difference between the central and parieto-occipital sites (Figure 1b). In contrast, in lows, all regions were different between each other (F > C-T, (*t*(14) = 10.99, *p* = 0.0001) and P-O (*t*(14) = 5.70, *p* = 0.0001), C-T < P-O (*t*(14) = 4.42, *p* = 0.001). 

Raw data for all regions in each experimental condition are reported in Appendix A.

#### 3.1.2. Alpha PSD

At hemispheric level, the alpha PSD exhibited a significant **Group** effect (*F*(1, 29) = 4.19, *p* = 0.037, η^2^ = 0.141, α = 0.560), with the highs’ value larger than lows’, and significant **Side × Group** (*F*(1, 29) = 9.30, *p* = 0.004, η^2^ = 0.255, α = 0.861) and Side × Region interactions (*F*(1, 58) = 21.47, *p* = 0.001, η^2^ = 0.427, α = 1.00). Decomposition of the former revealed a higher PSD on the left than on the right side in highs (*F*(1, 15) = 21.06, *p* = 0.001) and no side difference in lows. The latter interaction (Figure 2a) was sustained, in averaged groups, by a significantly larger PSD on both sides at the frontal than central-temporal regions (left, *t*(30) = 7.98, *p* = < 0.0001; right, *t*(30) = 20.49, *p* = < 0.0001) and at the central than posterior regions (left, *t*(30) = 7.97, *p* = < 0.0001; right, *t*(30) = 17.83, *p* < 0.0001). Moreover, the right PSD was lower than the left one at the central-temporal level (*t*(30) = 5.77, *p* = < 0.0001).

The midline alpha PSD also exhibited a significant **Group** effect (*F*(1, 29) = 5.25, *p* = 0.029, η^2^ = 0.153, α = 0.601) with a higher PSD in the highs than in the lows (Figure 2b). 

Raw data for all regions in each experimental condition are reported in Appendix A.

#### 3.1.3. Gamma PSD

ANOVA revealed a significant Side × Group interaction (*F*(1, 29) = 8.14, *p* = 0.008, η^2^ = 0.219, α = 0.712), whose decomposition (Figure 3a) indicated no Side difference in highs and a higher gamma PSD on the right than on the left side in lows (*F*(1, 14) = 8.34, *p* = 0.012). Moreover, we observed a significant Region × Group interaction (*F*(2, 58) = 13.46, *p* = 0.001, η^2^ = 0.317, α = 0.942) with the highs’ PSD lower than lows’ (*t*(29) = 3.43, *p* = 0.021) in the parieto-occipital region (Figure 3b).

Decomposition of the significant Group × Region × Condition interaction (*F*(10, 290) = 2.23, *p* = 0.047, η^2^ = 0.071, α = 0.735) revealed a significant Region effect in highs (*F*(2, 30) = 120.44, *p* = 0.001), whose regions were all different between each other (F > P-O, *t*(15) = 7.17, *p* < 0.0001; P-O > C-T, *t*(15) = 6.39, *p* = 0.0001; F > C-T, *t*(15) = 26.83, *p* = 0.0001), and a significant Region effect (*F*(2, 28) = 85.44, *p* = 0.001) and Region × Condition interaction in lows (*F*(10, 140) = 4.64, *p* = 0.006). Decomposition of the latter revealed a higher gamma PSD in the frontal than central-temporal (except Post) and parieto-occipital regions (except Rce and NH), and no difference between the central-temporal and parieto-occipital PSD in all conditions. (Appendix A).

Decomposition of the significant Side × Region interaction (*F*(2, 58) = 11.46, *p* = 0.0001, η^2^ = 0.283, α = 0.958) showed a higher PSD in the F than C-T and P-O regions on both sides, a higher PSD in the P-O than C-T only on the left side, and a left PSD lower than the right PSD at the P-O level (Appendix A). 

At midline sites, we observed only a significant Region effect (*F*(2, 58) = 125.40, *p* = 0.0001, η^2^ = 0.812, α = 0.999), with Cz > Fz (*t*(30) = 7.88, *p* = 0.0001), Cz > Pz (*t*(30) = 13.85, *p* = 0.0001), Fz > Pz (*t*(30) = 15.19, *p* = 0.0001). 

Raw data for all regions in each experimental condition are reported in Appendix A.

### 3.2. EEG Recurrence Quantification Analysis 

Figure 4 illustrates DET and ENT mean values of highs and lows in different conditions.

The SHSS score correlated with DET during NH (ρ = 0.431, *p* = 0.016) and Post (ρ = 0.368, *p* = 0.041) and did not correlate with ENT. Thus, the cognitive processing of highs and lows across the session seems to be different, as shown in Figure 5 which illustrates the mRP of a low and a high hypnotizable subject during closed-eyes relaxation and hypnosis.

To support the relevance of the hypnotizability trait in the mRP differences across the session, we correlated the values of DET and ENT observed during hypnosis with those found during closed-eyes relaxation, which is the condition most similar to NH except for hypnotic induction. We observed, in fact, a significant positive correlation between R_CE_ and NH (DET, ρ = 0.646, *p* = 0.0001; ENT, ρ = 0.910, *p* = 0.0001). 

## 4. Discussion

Several studies described EEG correlates of hypnosis, but their results were often inconsistent owing to methodological differences [9,17,18,19,46]. The aim of the present study was to detect the role of formal hypnotic induction in the EEG activity of highs that are aware of their hypnotic score and waiting for hypnotic induction. Our study provides evidence against a major role of hypnotic induction in the EEG correlates of hypnosis despite the highs’ experience of changes in their state of consciousness.

In the frequency domain, we limited our investigation to theta, alpha and gamma PSD. Theta and alpha are, in fact, the EEG frequency bands most studied during relaxation and hypnosis, although inconsistent findings of their hypnotizability-related and hypnosis-induced difference have been reported [9,46], and high gamma activity has been considered a marker of high hypnotizability and hypnosis [9]. 

Present findings indicate that formal hypnotic induction did not influence the EEG spectral correlates of neutral hypnosis. In fact, lows exhibited a larger theta and gamma PSD on the right than on the left side, whereas highs did not exhibit any hemispheric difference, suggesting cognitive processing different from the highs from the very beginning of the session. 

The stability of theta, alpha and gamma PSD observed in both groups across the session suggests greater relevance of the expectation of hypnosis (likely sustained by different hypnotic level) rather than induction to the experience of trance in highs, while in lows, it accords with their scarce change in the state of consciousness.

The lower theta PSD exhibited by highs on the frontal midline with respect to lows may indicate lower cognitive effort, as the frontal-midline theta increases during cognitive tasks [47]. The theta source has been localized on the anterior cingulate and medial prefrontal cortices [48,49,50], both parts of the Default Mode Network, which is less activated during hypnosis [7,51]. 

The highs’ larger alpha PSD on the midline sites accords with a reduced activation of the Default Mode Network [8]. Higher alpha power has been associated with internally directed attention [52], which is likely to occur in highs during and/or when expecting hypnosis. Moreover, it has been considered responsible for top–down inhibitory processes for the exclusion of irrelevant input [53,54], as occurs in highs during hypnotic state/induction [1]. Finally, the alpha PSD, which did not differ between the right and left side in lows, was larger on the left than on the right side in highs throughout the session, according with the observation of pre-eminent right side activation during hypnosis in highs [55].

Our findings of gamma PSD do not accord with earlier reports of increased PSD over both hemispheres in the early hypnotic induction, decreased activity in the left and increased activity in the right hemisphere in the late hypnotic induction in highs, and reduction of the gamma PSD in lows in both hemispheres [9].

The findings cannot be compared to the EEG activities recorded during a long-lasting simple relaxation session [56]—increasing alpha, theta and gamma absolute power throughout the session—as in that case, alpha and theta were divided into sub-bands, the instructions for relaxation were repeated every 3 min and the cortical activities were measured by averaging the EEG activity all over the brain.

The method we used to characterize the EEG Recurrence Plot [42] provides a multidimensional description of the brain activity considering the simultaneous recording from all electrodes. This prevented the assessment of hemispheric/regional differences, although other measures of the EEG complexity did not reveal hemispheric differences [13]. We showed that the cognitive processes developing in highs and lows across the experimental session were qualitatively different, and that those occurring during hypnosis were closely related to those observed during closed-eyes relaxation. Such a finding should be supported, however, by the investigation of further mRP dimensions in a larger sample. In not-hypnotized highs [44,45], the original RQA method [40] revealed higher Determinism compared to lows on the midline parietal site (Pz) during the earliest 3 min of simple relaxation, which roughly corresponds to the present closed-eyes relaxation condition. They were followed by progressive increases in the lows’ DET, likely depending on relaxation, so that in the following 9 min there was no significant group difference. Those participants were aware of their hypnotizability score but had been explicitly told that no hypnotic induction will be administered. In the present study, the open-eyes relaxation preceding the closed-eyes condition may have buffered the difference observed during closed-eyes relaxation in the participants not expecting hypnosis. 

## 5. Limitations

The episodic experience of relaxation/meditation reported by a few participants may have biased the results. Other limitations are that our protocol did not include the direct assessment of the participants’ changes in the state of consciousness [2] during/soon after each experimental condition, and that a preliminary assessment of the expected changes was not performed [33,34]. We are aware that, in the absence of a direct assessment of expectations, the pre-eminent role of the expectation of hypnosis and hypnotizability over hypnotic induction in the highs’ experience of trance can only be inferred and that the reported changes in the state of consciousness over the entire session could be inaccurate. Nonetheless, we intentionally chose to prevent expectation-induced (placebo) responses and to allow the participants’ state of consciousness to flow freely throughout the session without interruptions. Moreover, spectral analysis may have failed to detect task-related differences in highs owing to their largely distributed and less-pronounced local changes in brain activity during sensory, motor and cognitive tasks [57,58,59]. Finally, we must consider that the effect sizes of the non-significant group × conditions interactions were quite low (below 0.1); thus, investigation in a larger sample is mandatory.

From a methodological point of view, present findings, together with current evidence, suggest that multidisciplinary (psychological, neurophysiological) and multiparametric approaches (power spectra, coherence, functional connectivity, topology) may assist in the characterization of the states of consciousness. Further improvement of the related research may be obtained by enrolling medium hypnotizable participants [60].

## 6. Conclusions

The main findings of the present study were that the power spectrum density of alpha, theta and gamma bands does not support the relevance of the hypnotic induction to the highs’ experience of hypnosis. For lows, who reported negligible changes in their state of consciousness despite a few changes in the EEG, a strategy of disengagement from the task can be hypothesized. The different regional power observed in highs and lows, however, may indicate different modes of information processing in the two groups. These results are consistent with earlier reports not showing significant differences between conditions in the parasympathetic heart rate variability of both groups—as measured by the root mean square of successive differences between normal heartbeats (RMSSD, ms)—whose increase is associated with heightened experience of altered state of consciousness [61]. The multidimensional Recurrence Plot, however, highlighted qualitative differences in the cognitive processes of highs and lows and a close relation between awake (R_CE_) and post-induction conditions (NH).

The theoretical significance of the described findings is in their support for socio-cognitive theories of hypnosis, in that they provide EEG evidence in line with behavioral/experiential findings indicating a major role of expectation and hypnotizability in the experience of hypnosis [23]. Moreover, the findings highlight the relevance of the experimental setting in hypnotizability/hypnosis-related research.

Despite the apparent negligible role of formal induction in our experimental condition, we must underline that the induction procedure may have greater importance in clinical contexts. Specifically, it could reinforce the interpersonal relation between the hypnotist and the subject though gestures, posture, tone and rhythm of voice, choice of words and grammar construction [62], which can modulate the activation of the right orbitofrontal cortex and central oxytocin system [63].

## Figures and Tables

**Figure 1 brainsci-13-00875-f001:**
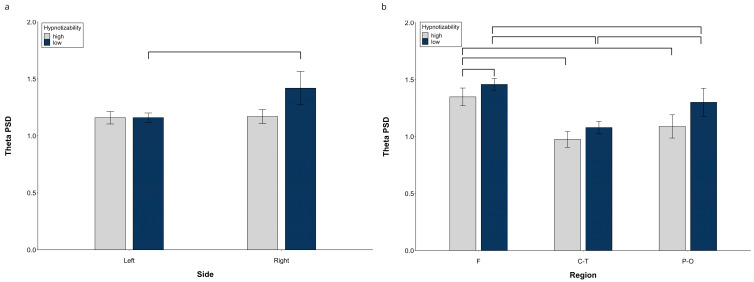
Theta PSD, raw data. (**a**) Hemispheric Side × Group interaction, (**b**) Midline Region × Group interaction. Lines indicate significant differences.

**Figure 2 brainsci-13-00875-f002:**
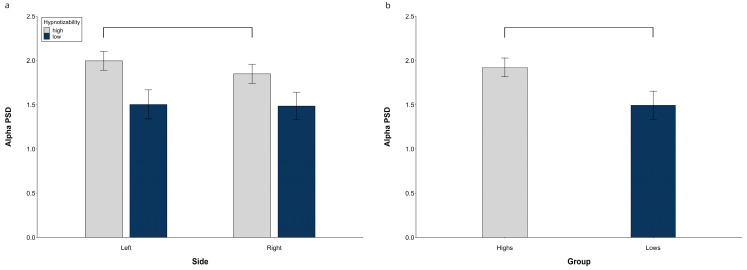
Alpha PSD, raw data. (**a**) Hemispheric Side × Group interaction, (**b**) Midline group difference. Lines indicate significant differences.

**Figure 3 brainsci-13-00875-f003:**
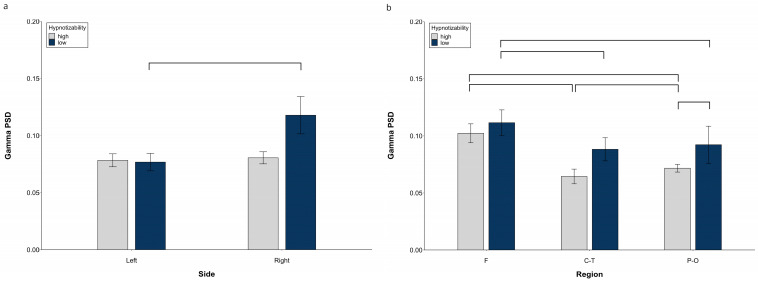
Gamma PSD, raw data. (**a**) Hemispheric Side × Group interaction, (**b**) Hemispheric Region × Group interaction. Lines indicate significant differences.

**Figure 4 brainsci-13-00875-f004:**
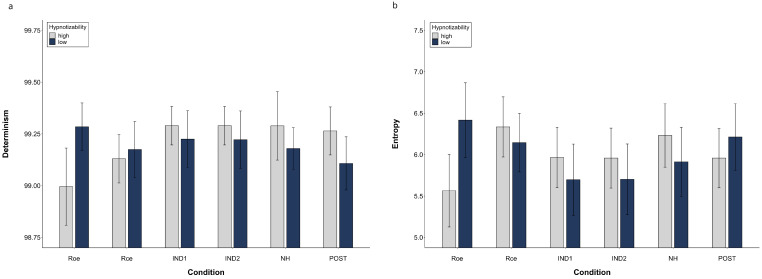
Determinism (**a**) and Entropy (**b**) of the Multidimensional Recurrence Plot. Raw data (mean, SEM).

**Figure 5 brainsci-13-00875-f005:**
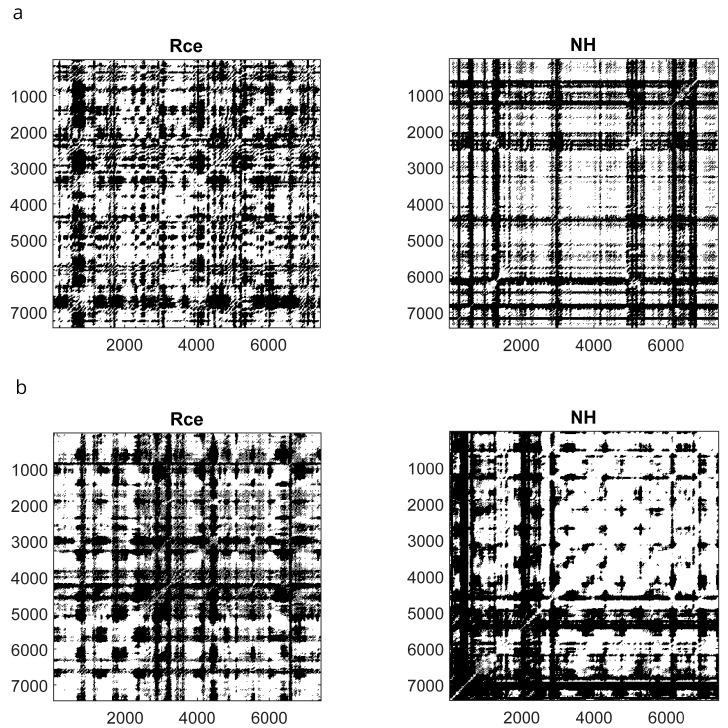
Recurrence Plots of a low ((**a**) upper panels) and a high ((**b**) lower panels) hypnotizable participant. Rce, closed eyes relaxation; NH, neutral hypnosis.

## Data Availability

The data are available upon request after the paper acceptance.

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
