# Peer review of "Is Hypnotic Induction Necessary to Experience Hypnosis and Responsible for Changes in Brain Activity?"

_brainsci, 2023, doi:10.3390/brainsci13060875_

Round 1
Reviewer 1 Report
Thank you for permitting me to review this manuscript
Please rephraze the primary objective of the study starting may be by reversing the order of the 2sentences
Please integrate the following recent articles probably related to this study
Higher hypnotic suggestibility is associated with the lower EEG signal variability in theta, alpha, and beta frequency bands Soheil KeshmiriID1 *, Maryam AlimardaniID1,2, Masahiro Shiomi1 , Hidenobu Sumioka1 , Hiroshi Ishiguro1,3, K PLOS ONE | https://doi.org/10.1371/journal.pone.0230853 April 9, 2020
Segregated brain state during hypnosis Jarno Tuominen 1,2,*, Sakari Kallio1,3, Valtteri Kaasinen2,4,5 and Henry Railo1,2,5,6 Neuroscience of Consciousness, 2021, 7(1): niab002 doi: 10.1093/nc/niab002
Bis spectrtal index appears also reliable in this context , it is easY to use please explain the choice of entropy in comparison to BIS
Please reconfirm the statistical data, I understand that 2 x 16 patients could possibly detect a satistical significant difference if there was big difference , but was the number enough to state that there is no difference in case of minimum defiiference , or in the simple terms is the the power of this study enough to draw conclusion ? especially when discussion starts with that most of other studies lacked power !
PLease suggest a future pathway for research to better describe controversies
Author Response
Rev 1
Thank you for permitting me to review this manuscript
We thank a lot this reviewer because her/his suggestion did help us to improve our manuscript
Please rephraze the primary objective of the study starting may be by reversing the order of the 2sentences
This change has been done in the revised version of the manuscript
Please integrate the following recent articles probably related to this study
- Higher hypnotic suggestibility is associated with the lower EEG signal variability in theta, alpha, and beta frequency bands Soheil KeshmiriID1 *, Maryam AlimardaniID1,2, Masahiro Shiomi1 , Hidenobu Sumioka1 , Hiroshi Ishiguro1,3, K PLOS ONE | https://doi.org/10.1371/journal.pone.0230853 April 9, 2020
- Segregated brain state during hypnosis Jarno Tuominen 1,2,*, Sakari Kallio1,3, Valtteri Kaasinen2,4,5 and Henry Railo1,2,5,6 Neuroscience of Consciousness, 2021, 7(1): niab002 doi: 10.1093/nc/niab002
Again, thanks for this suggestion. New sentences have been included in the Introduction owing to information provided by these papers, which have been included in the reference list
Bis spectrtal index appears also reliable in this context , it is easY to use please explain the choice of entropy in comparison to BIS
Despite some recent findings concerning the use of BIS to evaluate hypnosis depth (Dunham, C. M., Burger, A. J., Hileman, B. M., Chance, E. A., & Hutchinson, A. E. (2021). Bispectral Index Alterations and Associations With Autonomic Changes During Hypnosis in Trauma Center Researchers: Formative Evaluation Study. JMIR formative research, 5(5), e24044),
under the hypothesis that hypnosis represents a different but not reduced consciousness (as indicated by the reduced activity of the Default Mode Network) , we used DET and ENT of the Recurrence Plot to observe possible differences with respect to earlier findings obtained in not hypnotized participants (Madeo et al., 2013; Chiarucci et al., 2014) showing time related differences only in lows and differences between highs and lows only the very early minutes of the long lasting relaxation session.
Please reconfirm the statistical data, I understand that 2 x 16 patients could possibly detect a satistical significant difference if there was big difference , but was the number enough to state that there is no difference in case of minimum defiiference , or in the simple terms is the the power of this study enough to draw conclusion ? especially when discussion starts with that most of other studies lacked power !
We confirm statistics .Only the Group x Region x Condition interaction regarding gamma PSD showed a low effect size(η2 = .071, α = .735). Moreover, tables in the Suppl El Matshow that both significant and non significant differences were observed for the same number of subjects.
PLease suggest a future pathway for research to better describe controversies
From a methodological point of view the findings suggest that multidisciplinary (psychological, neurophysiological) and multiparametric approaches (power spectra, coherence, functional connectivity, topology) may assist in the characterization of the states of consciousness. Further improvement of the related research could be obtained by enrolling medium hypnotizable participants (Jensen et al., 2017)
Reviewer 2 Report
The study aimed to investigate the relationship between formal hypnotic induction, the experience of trance, and their neural correlates. It is uncertain whether formal hypnotic induction is significant in the experience of trance, as factors such as hypnotizability, beliefs, and expectations of hypnosis might have a more prominent role. The researchers assessed the EEG brain activity of participants with high and low hypnotizability scores who were aware of their hypnotizability level and informed about the session's inclusion of simple relaxation, formal hypnotic induction, and neutral hypnosis. Sixteen participants with high and low hypnotizability scores, determined by the Stanford Hypnotic Susceptibility Scale, form A, were included in the study. EEG recordings were obtained during consecutive conditions of open/closed eyes relaxation, hypnotic induction, neutral hypnosis, and post-hypnosis without interruptions for interviews. The variables of interest were theta, alpha, and gamma power spectrum density (PSD), as well as the Determinism (DET) and Entropy (ENT) of the EEG signal Multidimensional Recurrence Plot (mRP). The results showed that the highs reported more significant changes in their state of consciousness compared to the lows throughout the session. However, there were no condition-related changes in theta, alpha, and gamma PSD in both groups. Alpha PSD was higher in highs than in lows on midline sites, and the two groups exhibited different theta and gamma PSD patterns on different sides/regions, regardless of the conditions. ENT did not show any correlation with hypnotizability, while DET positively correlated with hypnotizability during hypnosis.
Overall, the findings suggest that the relevance of formal hypnotic induction to the experience of trance may be limited in individuals with high hypnotizability scores who are aware of their scores and have expectations of hypnosis. However, the mRP analysis indicated that cognitive processing varies depending on the level of hypnotizability.
I am very positive about studying altered states of consciousness. However, there are certain points in the following manuscript that concern me. Although the work has several strong points, there are some areas that require improvement. Firstly, the title is very brief and does not fully reflect the content. Similarly, the introduction needs further revision to provide a more comprehensive overview of the state of the art. In the methodology section, the information about the stimulus presentation software is not clear, and this should be explained. Additionally, more information is needed regarding the recording system and noise treatment. Lastly, in terms of the discussion and conclusions, it is necessary to highlight their practical and theoretical implications.
Author Response
Rev 2
The study aimed to investigate the relationship between formal hypnotic induction, the experience of trance, and their neural correlates. It is uncertain whether formal hypnotic induction is significant in the experience of trance, as factors such as hypnotizability, beliefs, and expectations of hypnosis might have a more prominent role. The researchers assessed the EEG brain activity of participants with high and low hypnotizability scores who were aware of their hypnotizability level and informed about the session's inclusion of simple relaxation, formal hypnotic induction, and neutral hypnosis. Sixteen participants with high and low hypnotizability scores, determined by the Stanford Hypnotic Susceptibility Scale, form A, were included in the study. EEG recordings were obtained during consecutive conditions of open/closed eyes relaxation, hypnotic induction, neutral hypnosis, and post-hypnosis without interruptions for interviews. The variables of interest were theta, alpha, and gamma power spectrum density (PSD), as well as the Determinism (DET) and Entropy (ENT) of the EEG signal Multidimensional Recurrence Plot (mRP). The results showed that the highs reported more significant changes in their state of consciousness compared to the lows throughout the session. However, there were no condition-related changes in theta, alpha, and gamma PSD in both groups. Alpha PSD was higher in highs than in lows on midline sites, and the two groups exhibited different theta and gamma PSD patterns on different sides/regions, regardless of the conditions. ENT did not show any correlation with hypnotizability, while DET positively correlated with hypnotizability during hypnosis.
Overall, the findings suggest that the relevance of formal hypnotic induction to the experience of trance may be limited in individuals with high hypnotizability scores who are aware of their scores and have expectations of hypnosis. However, the mRP analysis indicated that cognitive processing varies depending on the level of hypnotizability.
I am very positive about studying altered states of consciousness. However, there are certain points in the following manuscript that concern me. Although the work has several strong points, there are some areas that require improvement.
Firstly, the title is very brief and does not fully reflect the content.
The title has been changed
Similarly, the introduction needs further revision to provide a more comprehensive overview of the state of the art.
We have introduced a few recent papers indicating the role of dynamic approaches to the study of EEG. The review by Santarcangelo and De Pascalis 2020 reports the findings of several studies
In the methodology section, the information about the stimulus presentation software is not clear, and this should be explained.
We have explained that an experimenter verbally invited participants to relax, close eyes, listen to the prerecorded voice, open eyes.
Additionally, more information is needed regarding the recording system and noise treatment.
We specified which steps were performed to remove noise and artefacts from the EEG signal. Additionally, we emphasized that QA is particularly suited for this kinf of analysis as ot does not make any assumption on the stationarity, length, and noise of the time series
Lastly, in terms of the discussion and conclusions, it is necessary to highlight their practical and theoretical implications.
The theoretical significance of the described findings is their support to socio-cognitive theories of hypnosis, in that they provide EEG evidence in line with behavioral /experiential findings indicating a major role of expectation and hypnotizability in the experience of hypnosis (Lynn et al., 2015) Moreover, the findings highlight the relevance of the experimental setting in hypnotizability/hypnosis related research
Reviewer 3 Report
This is a very interesting paper investigating the EEG brain activity in patients receiving hypnotic induction. The paper is well written and of interest for the journal. However, several minor changes should be made before considering it for publication.
ABSTRACT
The main aims of the paper should be better clarified in the abstract section. I recommend to rephrase the aims according to the title.
The conclusions of the abstract should not be started with "The findings suggest"., and should not be a summary of the results or a repetition of results.
INTRODUCTION
I recommend to add some words about why is hypnosis important, and some words about the history or historical perspective of hypnosis.
I recommend to add more references about the neurobiological basis of hypnosis. Can it be linked with neurotransmitter systems? Structural and functional brain bases should be discussed in the introduction section
METHODS
The first part should be renamed as "Participants and study design" instead of "Subjects".
I recommend to add a brief Table explaining the variables used in the "2.4. Variables" subsection.
RESULTS
The main characteristics and description of the sample is lacking. I recommend to describe them.
DISCUSSION
Limitations and conclusions section should be divided into two sections. I recommend to draw a section about Limitations and strenghts, and a second section about conclusions. The conclusions section cannot contain references. They should be moved to the discussion ection.
Future perspective for further studies are needed.
Author Response
Rev 3
This is a very interesting paper investigating the EEG brain activity in patients receiving hypnotic induction. The paper is well written and of interest for the journal. However, several minor changes should be made before considering it for publication.
ABSTRACT
The main aims of the paper should be better clarified in the abstract section. I recommend to rephrase the aims according to the title.
The title has been changed according to the comment of rev 1
The conclusions of the abstract should not be started with "The findings suggest"., and should not be a summary of the results or a repetition of results.
The final sentences have been changed. The summary of the results has been left to enable us the understanding of the general significance of the study
INTRODUCTION
I recommend to add some words about why is hypnosis important, and some words about the history or historical perspective of hypnosis.
I recommend to add more references about the neurobiological basis of hypnosis. Can it be linked with neurotransmitter systems? Structural and functional brain bases should be discussed in the introduction section
The main differences observed by imaging studies between hypnosis and wakefulness, already reported, have been explained .
Cardeña, E.; Jönsson, P.; Terhune, D. B.; Marcusson-Clavertz, D. The Neurophenomenology of Neutral Hypnosis. Cortex 2013, 49 (2), 375–385. https://doi.org/10.1016/j.cortex.2012.04.001.
- Halligan, P. W.; Oakley, D. A. Hypnosis and Cognitive Neuroscience: Bridging the Gap. Cortex 2013, 49 (2), 359–364. https://doi.org/10.1016/j.cortex.2012.12.002.
- Jiang, H.; White, M. P.; Greicius, M. D.; Waelde, L. C.; Spiegel, D. Brain Activity and Functional Connectivity Associated with Hypnosis. Cortex 2016, cercor;bhw220v1. https://doi.org/10.1093/cercor/bhw220.
- Landry, M.; Lifshitz, M.; Raz, A. Brain Correlates of Hypnosis: A Systematic Review and Meta-Analytic Exploration. Neuroscience & Biobehavioral Reviews 2017, 81, 75–98. https://doi.org/10.1016/j.neubiorev.2017.02.020.
Neurotransmitters has been associated with hypnotizability rather than hypnosis, despite the title of recent reviews:
Cardeña, E.; Jönsson, P.; Terhune, D. B.; Marcusson-Clavertz, D. The Neurophenomenology of Neutral Hypnosis. Cortex 2013, 49 (2), 375–385. https://doi.org/10.1016/j.cortex.2012.04.001.
METHODS
The first part should be renamed as "Participants and study design" instead of "Subjects".
We have accepted this comment, but we are not sure that it is in line with the journal guidelines
We interpreted the reviewer’s request as an invitation to briefly indicate the reason why the studied variables were chosen.. Thus, we included references (all cited in the introduction) in the Variables sub-section.
RESULTS
The main characteristics and description of the sample is lacking. I recommend to describe them.
The characteristics of the sample are present in the Subljects paragraph (gender, age, hypnotizability, education -University students)
DISCUSSION
Limitations and conclusions section should be divided into two sections. I recommend to draw a section about Limitations and strenghts, and a second section about conclusions.
This is not required by the journal. Nonetheless, we accepted the reviewer’s comment
The conclusions section cannot contain references. They should be moved to the discussion section.
This is not in line with the journal guidelines. We think not to be allowed to cancel references
Future perspective for further studies are needed.
In line with the requests of Rev 1 and Rev 2, a few lines regarding this aspect of the study have been included
Lines 363-368
From a methodological point of view present findings together with current evidence suggest that multidisciplinary (psychological, neurophysiological) and multiparametric approaches (power spectra, coherence, functional connectivity, topology) may assist in the characterization of the states of consciousness. Further improvement of the related research could be obtained by enrolling medium hypnotizable participants (Jensen et al., 2017).
Line 383-387
The theoretical significance of the described findings is their support to socio-cognitive theories of hypnosis, in that they provide EEG evidence in line with behavioral /experiential findings indicating a major role of expectation and hypnotizability in the experience of hypnosis (Lynn, et al., 2015). Moreover, the findings highlight the relevance of the experimental setting in hypnotizability/hypnosis related research
Round 2
Reviewer 2 Report
Thank you for addressing my comments